# Self-construal and attentional biases in cognitive processing: Insights from Chinese college students for mental health contexts

Jing Li[1]*, Sijia Liu[2], Hongsheng Peng[3], Liwu Tang[1], Lin Yuan[4]

1 School of Public Administration, Chongqing Technology and Business University, Chongqing, China,
2 School of Marxism, Chongqing University of Education, Chongqing, China, 3 Faculty of Life Sciences and Medicine, King's College London, London, United Kingdom, 4 School of Teacher Education, Southwest University, Chongqing, China

* lijing75@ctbu.edu.cn

## Abstract

This study integrates Markus and Kitayama's self-construal theory with the Biopsychosocial Model to examine the effects of self-construal on cognitive biases and their implications for mental health among Chinese college students. It investigates how independent (IndSC) and interdependent (InterSC) self-construals influence cognitive biases towards positive and negative stimuli, emphasizing the mediating roles of attentional control, self-esteem, cognitive reappraisal, and the need to belong. The research utilizes a hybrid sampling strategy, engaging 748 undergraduate students from diverse majors and universities in Chongqing, China. Participants completed assessments measuring self-construal, attentional control, self-esteem, cognitive reappraisal, need to belong, and cognitive biases. Employing structural equation modeling, the study uncovers that IndSC is positively associated with cognitive biases, favoring positive over negative information, with attentional control and self-esteem serving as significant mediators. InterSC, however, promotes a balanced attention to both positive and negative information, with cognitive reappraisal and the need to belong acting as mediators. These findings underscore the significant role of self-construal within the broader biopsychosocial framework in shaping cognitive processes and highlight the importance of considering these factors in mental health interventions. By leveraging a comprehensive sampling approach, the study ensures a representative sample, enhancing the generalizability of its findings to the broader Chinese college student population. This research not only contributes to the understanding of the complex interplay between cultural psychology and mental health but also suggests the need for mental health interventions that are culturally and individually tailored. Future research should extend to other cultural settings and employ longitudinal designs to further explore the dynamic interactions between self-construal, cognitive biases, and mental health from a biopsychosocial perspective.

**Data Availability Statement:** Our database, containing the original data associated with our research paper, has been made publicly available for future research and verification purposes. It can

be accessed via the following Digital Object Identifier (DOI): 10.6084/m9.figshare.25061078.

**Funding:** This work was supported by Humanities and Social Sciences Research General Project, Ministry of Education of the People's Republic of China (21XJA190002 to JL). The funder had no role in study design, data collection and analysis, decision to publish, or preparation of the manuscript.

**Competing interests:** The authors have declared that no competing interests exist.

# 1. Introduction

Recent research has highlighted that cognitive biases towards attentional focus on positive versus negative information can serve as risk or protective factors for mental health outcomes. For instance, an enhanced focus on positive information has been associated with higher levels of positive affectivity [1], post-traumatic growth [2] and hope [3], while a preoccupation with negative stimuli is often linked to increased vulnerability to Social Anxiety Disorder [4], anxiety trajectories [5], depression [6,7], negative affectivity [1], psychological distress [3], emotional vulnerability [8], and post-traumatic stress disorder [2]. The concept of self-construal, as proposed by Markus and Kitayama [9], has profound implications for understanding individual psychological functioning within cultural contexts. In the domain of mental health, exploring the impact of self-construal on cognitive biases is particularly salient, as these biases are closely linked to emotional health and psychopathology. Independent self-construal (IndSC) and interdependent self-construal (InterSC) are thought to influence how individuals process information and regulate emotions [9], which are critical components in the development and maintenance of mental health disorders, especially among college students who are navigating complex social and academic environments. This association underscores the importance of understanding the mechanisms through which self-construal can shape cognitive biases and, consequently, influence mental well-being.

While traditionally explored in a cross-cultural context [9], we focus on the individual level of self-construal, as suggested by Gabriel et al. [10] and Chiu et al. [11]. This approach minimizes confounding variables often present in broader cultural comparisons [12] and is particularly relevant in the Chinese context, where traditional collectivism intersects with modern individualistic influences.

This study is grounded in Markus and Kitayama's theory of self-construal and further employs the Biopsychosocial Model [13]. The Biopsychosocial Model emphasizes the integrated role of biological, psychological, and social factors in shaping human behavior and mental health. Within this framework, we explore how self-construal (a social factor), attention control (a cognitive factor), cognitive reappraisal (an emotional regulation strategy), self-esteem (a personality trait), and the need to belong (an affective motivation) collectively influence an individual's cognitive biases. This study aims to delve into these mechanisms within the Chinese cultural milieu, where collective values significantly impact individual psychology. Given the critical transition period that college represents in the lifespan, understanding how IndSC and InterSC impact cognitive biases among Chinese college students can inform culturally sensitive mental health interventions designed to promote resilience and emotional well-being.

# 2. Literature review and hypothesis development

## 2.1 The interface of self-construal and cognitive biases in mental health

The exploration of self-construal's impact on cognitive biases is vital for cultural psychology and its intersections with mental health research. Cognitive biases, such as preferential processing of certain stimuli [14,15], are not only pivotal in understanding individual psychological functioning but also in identifying risk factors for mental health conditions [16,17]. While the existence of cross-cultural variations in thinking styles is well-documented [18], the specific manifestation of cognitive biases across cultures [19], particularly concerning mental health, requires further investigation. Moreover, most previous research exploring the influencing factors of attentional cognitive bias has focused on genetic markers [20] and cognitive mechanisms [8]. Existing studies show mixed results regarding the effects of independent (IndSC) and interdependent (InterSC) self-construal on cognitive biases [21,22]. In light of

this, our study aims to illuminate how independent (IndSC) and interdependent (InterSC) self-construals shape cognitive biases within a mental health framework, focusing on the Chinese cultural context.

In light of our investigation into the interplay between self-construal and cognitive biases within the realm of mental health, we posit the following hypotheses tailored to the Chinese cultural context:

H1a: IndSC among Chinese college students will be positively correlated with attention to positive information (API) and inversely correlated with attention to negative information (ANI). This hypothesis anticipates that a more individualistic orientation will engender a cognitive bias that favors positive over negative stimuli, potentially serving as a buffer against mental health challenges.

H1b: Conversely, InterSC will be inversely related to API and positively related to ANI. This suggests that a collectivist orientation may predispose individuals to a cognitive bias that places undue focus on negative stimuli, which could be a potential risk factor for adverse mental health outcomes.

## 2.2 Mediating mechanisms: Cognition, motivation, and emotion regulation

### 2.2.1 Attentional control as the potential mediator of self-construal on cognitive bias.
Attentional control, an essential element of executive functioning, orchestrates our focus, perception, and cognitive processing, thereby playing a critical role in mental health outcomes. It is the cognitive ability to manage what we pay attention to and what we ignore, with significant implications for our emotional well-being [23].

Emerging research underscores the impact of self-construal on attentional processes. For instance, individuals with a predominantly InterSC are found to have a broader visual attention field, suggesting a more comprehensive scanning of the social environment [24]. This expansive attentional scope may be adaptive in collectivist cultures where social harmony is paramount but could also contribute to heightened sensitivity to social threats, which in turn may influence mental health [25]. Furthermore, attention to interpersonal context, a characteristic of InterSC, can impede the completion of tasks that require focus away from social cues [26].

Given the association between attentional biases and mental health disorders such as anxiety and depression, understanding the mediating role of attentional control within the self-construal framework can inform targeted mental health interventions. We hypothesize that attentional control mediates the relationship between self-construal and cognitive biases, potentially offering a pathway to modify attentional biases in mental health contexts.

Attention control is also a function of working memory that adjusts attentional resources in response to task demands [23]. Its role in regulating responses to threat-related information [27] is particularly relevant to mental health, as it can either contribute to stress and anxiety or protect against them. Cognitive bias modification research reveals that bolstering attentional control can recalibrate cognitive biases [8,28]. Interestingly, changes in attentional biases following modification procedures appear to occur independently of changes in attentional control [29].

Therefore, we propose a nuanced cognitive mechanism whereby self-construal influences cognitive biases via attentional control. We hypothesize:

H2a: IndSC will be positively associated with attentional control. In turn, enhanced attentional control will correlate with greater attention to positive information (API) and reduced attention to negative information (ANI), reflecting a protective cognitive bias conducive to positive mental health outcomes.

H2b: Conversely, InterSC will be negatively associated with attentional control. This diminished attentional control is expected to be associated with less attention to positive information (API) and greater attention to negative information (ANI), potentially indicating a cognitive bias that may be a risk factor for negative mental health outcomes.

**2.2.2 Motivation as the potential mediator of self-construal on cognitive bias.** *Self-esteem as the Potential Mediator of Self-Construal on Cognitive Bias*

The interplay between self-construal and self-esteem is a critical factor in cognitive bias formation. Individuals with a high IndSC tend to report higher self-esteem, focusing on self-worth and personal accomplishments [30–32]. This tendency is reflected in self-reports where independents often rate their self-esteem more positively than interdependents, although this difference may not be as pronounced in implicit self-esteem measures [33]. High self-esteem is correlated with greater receptivity to positive social feedback and openness to social interactions [34]. Therefore, this heightened self-esteem in IndSC individuals is associated with a cognitive bias toward positive information [35], fostering a protective mental health environment.

Conversely, InterSC is often correlated with lower self-esteem, due in part to a focus on social harmony and adaptability rather than personal achievement [9,33]. For example, Chinese university students demonstrated a reliance on competence self-esteem under independent priming conditions, while interdependent priming conditions led to a preference for self-esteem based on social relationships [36]. InterSC was positively associated with relationship harmony; IndSC was positively correlated with self-esteem [37]. This variation in self-esteem might influence cognitive processing, leading InterSC individuals to pay more attention to negative information [35], potentially heightening vulnerability to mental health issues. This is evident in various cognitive and emotional responses, such as attentional biases towards rejection in individuals with low self-esteem [38] and faster processing of negative emotions [39].

Hypothesis H3a: IndSC will be positively associated with self-esteem, which in turn will be positively associated with API and negatively associated with ANI.

Hypothesis H3b: Conversely, InterSC will be negatively associated with self-esteem, which will lead to decreased API and increased ANI.

*Need to Belong as the Potential Mediator of Self-Construal on Cognitive Bias*

The need to belong is a fundamental human drive, with marked variations based on self-construal. influencing their cognitive processing [9]. Individuals with a strong InterSC typically prioritize maintaining harmonious relationships and seek to empathize and connect with others, as highlighted in various cultural studies [9,40,41]. Individuals influenced by InterSC values demonstrate a stronger motivation for acceptance, conformity, and communal cohesion [42,43]. This heightened need to belong in InterSC individuals may lead to a cognitive bias that emphasizes negative social stimuli. For instance, InterSC individuals may exhibit a cognitive bias that de-emphasizes self-enhancement in favor of maintaining social harmony [44]. This is reflected in a lonely individuals, with heightened sensitivity to social threats [45] and potential rejection [46]. Importantly, the need to belong is closely connected to loneliness, with those reporting a stronger need to belong also experiencing higher levels of loneliness [47], for example, students who exhibited a higher need to belong were also experiencing higher levels of loneliness [48].

On the other hand, individuals with a strong IndSC might exhibit lower levels of affiliation motivation [49,50], focusing more on individual achievements and personal attributes. This may lead to a cognitive bias towards positive stimuli and away from negative social stimuli.

Hypothesis H4a: IndSC will be negatively associated with the need to belong, which will be inversely related to API and positively related to ANI.

Hypothesis H4b: In contrast, InterSC will be positively associated with the need to belong, which will lead to a decrease in API and an increase in ANI.

### 2.2.3 Emotion regulation (cognitive reappraisal) as a mediator. *Self-Construal and Emotion Regulation*

Emotion regulation, a fundamental aspect of human psychology, encompasses strategies for managing and modulating emotional responses. Self-construal significantly influences how individuals perceive and regulate their emotions, as posited by Markus and Kitayama [9]. This study focuses on cognitive reappraisal, a strategy that entails reinterpreting emotional stimuli to alter emotional responses.

Empirical studies have shown that emotion regulation strategies systematically vary according to one's self-construal [9,51–53]. Those with a high InterSC often rely on strategies like expressive suppression to align their emotional responses with collective cultural norms [54–56]. In contrast, individuals with a predominant IndSC are more inclined towards strategies that foster positive emotional experiences, as typically observed in Western cultures [54,55,57]. The consequences of expressive suppression differ between individuals from independent and interdependent cultures, with negative mental health outcomes for the former and potential mental health benefits for the latter [58–61].

*Cognitive Reappraisal and Cognitive Bias*

Cognitive reappraisal, a crucial emotion regulation strategy, involves reshaping the interpretation of emotional stimuli, which can significantly alter emotional outcomes and attentional focus [62–64]. The effectiveness and application of cognitive reappraisal can differ across cultures, influencing the ways emotions are processed and expressed [65].

The relationship between cognitive reappraisal and cognitive biases is both complex and profound. By modifying the interpretation of emotional stimuli, cognitive reappraisal can significantly influence emotional responses, thus impacting cognitive biases [66–68]. Furthermore, cognitive biases have been linked to the use of emotion regulation strategies, with cognitive reappraisal reducing negative emotions by altering stimulus meaning and attention distribution [66], thereby mitigating negative attentional biases [69]. Research on individuals with social anxiety disorder also suggests that cognitive reappraisal is associated with positive interpretations of ambiguous social information [70]. This suggests that cognitive reappraisal could serve as a critical link between self-construal and the formation of cognitive biases.

Hypothesis H5a: We hypothesize that cognitive reappraisal mediates the relationship between self-construal and cognitive biases. Specifically, individuals with IndSC, who are inclined to utilize cognitive reappraisal, are expected to demonstrate a bias towards processing positive emotional stimuli and mitigating the impact of negative information.

Hypothesis H5b: In contrast, individuals with InterSC, who may lean more towards suppression strategies, are likely to exhibit cognitive biases that favor the processing of negative emotional stimuli, potentially due to their heightened focus on maintaining social harmony and avoiding conflict.

## 2.3 The current study

The current study addresses a gap in the existing research on cultural influences on attentional cognitive bias, particularly within the Chinese cultural context. Grounded in Markus and Kitayama's self-construal framework, our research explores how independent (IndSC) and interdependent (InterSC) self-construals affect cognitive biases, with a focus on mechanisms involving attentional control, self-esteem, need to belong, and cognitive reappraisal.

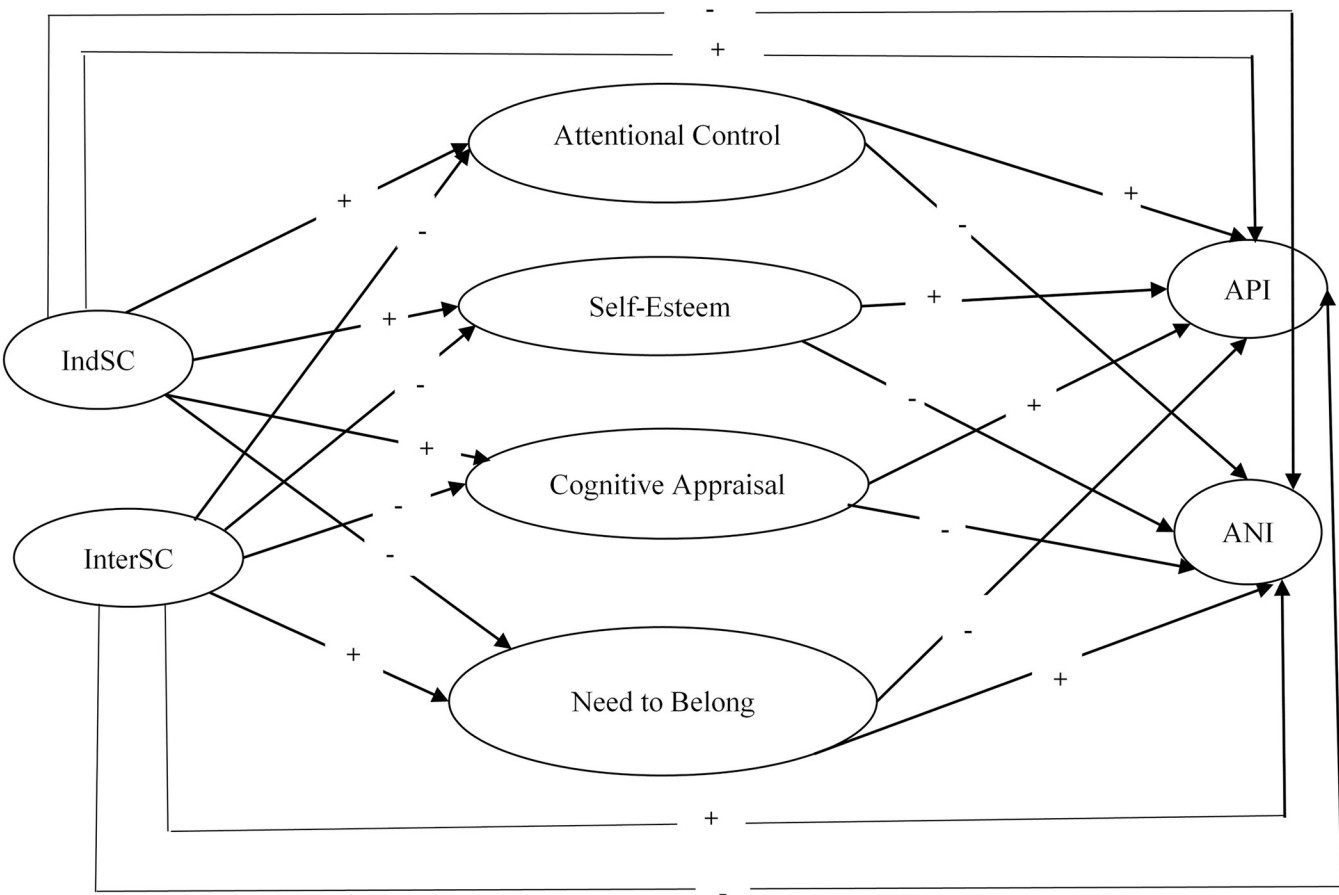

**Fig 1. Conceptual model: Attentional control, self-esteem, cognitive appraisal and need to belong as mediators in the relationship between self-construal and cognitive bias in Chinses college students.** *Note*: IndSC = interdependent self-construal; InterSC = interdependent self-construal; API = attention to positive information; ANI = attention to negative information.

We propose that self-construal exerts a significant influence on attentional cognitive biases through a series of cognitive, motivational, and emotional pathways. Specifically:

InterSC Hypothesis: InterSC is hypothesized to be negatively associated with attentional control, self-esteem, and cognitive reappraisal. These factors are anticipated to contribute to a cognitive bias that favors ANI and disfavors API.

IndSC Hypothesis: Conversely, IndSC is expected to show positive associations with attentional control, self-esteem, and cognitive reappraisal, leading to a cognitive bias that favors API and disfavors ANI.

Fig 1 in our paper visually encapsulates these hypotheses and their interrelationships, presenting a conceptual model that elucidates the proposed dynamics between self-construal, the mediating variables, and the resulting cognitive biases.

## 3. Methods

### 3.1 Participants and procedures

This study, focusing on the intersection of self-construal and cognitive biases in the context of mental health, employed a hybrid sampling strategy to gather data from a diverse and representative sample of college students in Chongqing, a major urban center in western China.

**3.1.1 Ethics statement.** Ethical approval for this study was granted by the Academic Ethics Committee of Chongqing Technology and Business University, adhering to rigorous ethical standards for conducting mental health research. Participants were thoroughly informed about the research objectives and their rights, including the freedom to withdraw at any time, before providing written informed consent.

A focused screening was implemented to rule out biases due to undiagnosed mental health conditions, utilizing a detailed questionnaire on participants' mental health history and symptoms, with an emphasis on anxiety and depression. This process ensured the cognitive biases identified were directly related to self-construal. Throughout the study, our commitment to ethical research safeguarded the participants' well-being and privacy.

**3.1.2 Sampling methodology.** Purposive Sampling: Chongqing was selected for its diverse socio-cultural milieu, providing a dynamic urban setting reflective of modern Chinese society.

Stratified Random Sampling:

University Selection: Participants were recruited from two institutions: Chongqing Technology and Business University, a comprehensive university, and Chongqing Second Normal College, a specialized professional university, to encompass a wide range of educational environments.

Major Selection: Various academic majors were randomly selected within these institutions to ensure disciplinary diversity.

Class Selection: Specific classes within these majors were randomly chosen to enhance sample representativeness.

Survey Subjects: All students in the selected classes were invited to participate, promoting inclusive participation.

Formal sample size and power calculations were not conducted prior to participant recruitment due to the exploratory nature of this study, which examines the nuanced relationship between self-construal and cognitive biases in mental health contexts. Nevertheless, the achieved sample size of 748 participants far exceeds typical requirements, ensuring substantial statistical power to identify significant findings across the study's analyses. The hybrid sampling strategy, encompassing a wide educational and disciplinary spectrum from two distinct types of universities in Chongqing, guarantees a diverse and representative sample. This approach enhances the reliability and generalizability of our findings, effectively compensating for the initial absence of sample size and power calculations.

**3.1.3 Participant recruitment.** The study was conducted from September 8, 2021, to November 30, 2021, involving 748 undergraduate students (243 males, 505 females) aged 19 to 22 years (Mean = 20.5, SD = 1.1). The sample included 396 students (52.9%) from the comprehensive university and 352 (47.1%) from the professional university, with participants mainly from Science and Engineering (271, 36.2%), Public Administration (212, 28.3%), Preschool Education (94, 12.6%), Business Administration (87, 11.6%), and Economics (84, 11.2%). The study involved students from various academic years, including sophomores (289, 38.6%), juniors (318, 42.5%), and seniors (141, 18.9%).

Recruitment was conducted via on-campus promotions and during mental health classes, emphasizing the study's relevance to mental health. All participants provided written informed consent, in accordance with the ethical standards of the institutional review board, prior to their inclusion in the study. They were assured of the confidentiality of their responses and the option to withdraw or omit questionnaire sections at any time. All data were anonymized to maintain participant confidentiality.

## 3.2 Measures

**3.2.1 Self-construal.** The 24-item Self-Construal Scale (SCS) by Singelis [71], offering separate scores for Independent Self-Construal (IndSC) and Interdependent Self-Construal (InterSC), was used. This scale is relevant to mental health research as it allows for the exploration of how individualistic and collectivistic orientations influence cognitive and emotional processes in the context of mental health. Participants rated their responses on a 7-point Likert-type scale, ranging from 1 (strongly disagree) to 7 (strongly agree). Reliability assessments in previous research have reported Cronbach's alpha values of .73 and .74 for the IndSC scale and .69 and .70 for the InterSC scale [71]. In our current sample, the IndSC ($\alpha$ = .66) and InterSC ($\alpha$ = .71) subscales exhibited commendable reliability.

**3.2.2 Attentional control.** The 20-item Attentional Control Scale (ACS) [72] assesses the ability to focus attention amidst distractions, a crucial aspect in understanding cognitive processing and its implications for mental health, particularly in stress and anxiety disorders. The total score provides an overall measure of attentional control, while two subscales, namely the shifting subscale (evaluating the ability to switch attention) and the focusing subscale (reflecting the capacity to concentrate in the presence of distraction), offer a more detailed assessment. Responses were recorded on a 4-point Likert scale (1 = almost never; 2 = sometimes; 3 = often; 4 = always). We employed the Chinese version of the ACS [73], which has exhibited robust reliability and validity in previous studies involving Chinese college students. Within our current sample, the total scale demonstrated a Cronbach's alpha of .79, with subscale reliabilities of .73 (focusing) and .67 (shifting).

**3.2.3 Self-esteem.** Rosenberg's Self-Esteem Scale [74], a widely accepted measure, was employed to evaluate self-esteem levels. Self-esteem is a significant factor in various mental health outcomes, including depression and anxiety, making it a vital component in our study. This scale consists of 10 items, each rated on a 4-point scale ranging from 1 (strongly disagree) to 4 (strongly agree), with higher scores indicating higher self-esteem. Reverse scoring was applied to appropriate items, and the scores from all items were aggregated to yield a total self-esteem score. The internal consistency of this scale within our current study was excellent, exhibiting a Cronbach's alpha coefficient of .88.

**3.2.4 Cognitive appraisal.** The 10-item Emotion Regulation Questionnaire (ERQ) [63], particularly the Reappraisal subscale, was used to measure cognitive reappraisal strategies. Given the importance of emotion regulation in mental health, understanding its role in cognitive processing is critical. Ratings were provided on a 7-point Likert scale, with responses reflecting typical experience and expression of emotions (from 1 = strongly disagree to 7 = strongly agree). Utilization of the ERQ within Asian youth populations has reaffirmed the original factorial structure of the instrument [75]. In our current sample, the Cronbach's alpha coefficient for this scale was excellent at .84.

**3.2.5 Need to belong.** The 10-item Need to Belong Scale [76] was utilized to assess participants' desire for social acceptance and group inclusion, factors that are closely linked to mental health issues such as social anxiety and loneliness. Responses on the Need to Belong scale were recorded on a 7-point scale ranging from "strongly disagree" (1) to "strongly agree" (7). The score of need to belong is formed by averaging the scores of all items, the higher the score, the higher the level of need to belong. Previous research by Pickett et al. [77] has employed this scale to examine sensitivity to social cues, showing sufficient reliability with a Cronbach's alpha coefficient of 0.83. In our present study, the scale exhibited a Cronbach's alpha of .79.

**3.2.6 Cognitive bias.** The 22-item Attention to Positive and Negative Information (APNI) scale [1], comprising the API and ANI subscales, was used to assess cognitive biases. This scale is particularly relevant for mental health research as it provides insights into how individuals

differentially process emotional information, a key aspect in understanding various psychological conditions. Participants provided responses on a five-point Likert scale, ranging from 1 (very untrue of me) to 5 (very true of me), indicating the degree to which they attend to positive and negative information. Importantly, the low correlation between attention to positive and negative information attests to the independence of these two constructs [1]. Due to its practicality, the APNI scale has been widely utilized since its inception [78,79]. For this study, we employed the Chinese version [80], which has demonstrated sufficient reliability and validity in Chinese college students. In our current sample, the Cronbach's coefficients for API and ANI were 0.80 and 0.76, respectively.

### 3.3 Statistical analysis

In our study, the statistical analysis was conducted with a comprehensive approach using SPSS 25.0 and Amos 24.0 software. Initially, we undertook reliability tests to ensure the validity and consistency of our data. This was followed by a common method variance test aimed at identifying any methodological biases that could influence the results.

The core of our analysis centered around testing the primary hypothesis, which posited mediation effects of attentional control, self-esteem, cognitive reappraisal, and the need to belong. To this end, structural equation modeling (SEM) was employed. The efficacy of our SEM approach was assessed through a series of fit indices: the Chi-square statistic provided a basic indication of model fit, while the Root Mean Square Error of Approximation (RMSEA) offered insights into how well the model approximated the real-world data. Additionally, Comparative Fit Index (CFI), Tucker-Lewis Index (TLI), and Incremental Fit Index (IFI) were utilized as comparative measures of model fit, comparing our proposed model against a baseline null model. A good model fit, as per our criteria, was indicated by RMSEA values being below .06, and CFI, TLI, and IFI values being equal to or greater than .95, aligning with standard thresholds in the field [81].

Further deepening our analysis, we conducted a partial mediation effect analysis. This was pivotal in detailing the mediation roles and involved employing bootstrapping methods for more accurate confidence interval estimation. This comprehensive approach was instrumental in dissecting the direct, indirect, and total effects, which are crucial for a nuanced understanding of the interplay between self-construal and cognitive biases.

## 4. Results

In the Results section, we first address the issue of common method variance (CMV). To manage this, we implemented procedural controls, including item reversals. Our application of Harman's one-factor test revealed that the largest factor accounted for only a modest 11.14% of the variance, which is significantly below the commonly accepted threshold of 40%. This result indicates minimal concerns regarding CMV in our data, thereby bolstering the credibility of our findings.

Subsequently, we delved into the statistical analysis results. Our initial focus was on descriptive statistics and the correlations among the study variables. These analyses were crucial for establishing the foundational relationships within our data set. In Table 1, we present a comprehensive descriptive and correlation analysis of the study variables, drawing from a sample of 748 Chinese college students. Initial analyses focused on descriptive statistics and correlations among study variables. IndSC showed positive correlations with API (p < .001), attentional control (p < .001), self-esteem (p < .001), and cognitive reappraisal (p < .001). Notably, no significant correlation was found between IndSC and ANI (p = .636) or the need to belong (p = .67). InterSC was positively correlated with API (p < .001), ANI (p < .001), cognitive

**Table 1. Descriptive and correlation analysis of the study variables ($n$ = 748).**

| Variables | 1 | 2 | 3 | 4 | 5 | 6 | 7 | 8 |
|---|---|---|---|---|---|---|---|---|
| 1. IndSC | 1 | | | | | | | |
| 2. InterSC | .336*** | 1 | | | | | | |
| 3. Attentional Control | .216*** | -.016 | 1 | | | | | |
| 4. Self-esteem | .353*** | .059 | .435*** | 1 | | | | |
| 5. Cognitive Appraisal | .277*** | .279*** | .152*** | .284*** | 1 | | | |
| 6. Need to Belong | -.016 | .216*** | -.308*** | -.127*** | .010 | 1 | | |
| 7. API | .361*** | .313*** | .133*** | .346*** | .369*** | .141*** | 1 | |
| 8. ANI | -.017 | .151*** | -.303*** | -.316*** | .082* | .348*** | .306*** | 1 |
| Mean | 57.66 | 59.70 | 40.75 | 28.67 | 31.62 | 33.19 | 40.84 | 37.55 |
| SD | 8.28 | 7.96 | 6.21 | 5.19 | 5.59 | 6.47 | 5.99 | 6.13 |

*Note*: API = attention to positive information; ANI = attention to negative information; IndSC = independent self-construal; InterSC = interdependent self-construal.

\* $P<.05$

\*\*\* $P<.001$.

reappraisal (p < .001), and the need to belong (p < .001), but not with attentional control (p = .655) or self-esteem(p = .108).

Additionally, API and ANI were variously correlated with the mediating variables. Correlations between API and attentional control (p < .001), self-esteem (p < .001), cognitive reappraisal (p < .001), and the need to belong (p < .001) were all positive, suggesting a bias toward positive emotional processing. Conversely, ANI showed negative correlations with attentional control (p < .001) and self-esteem (p < .001) but was positively associated with cognitive reappraisal (p < .05) and the need to belong (p < .001), indicating a bias towards negative emotional processing.

The crux of our results section, however, centered on exploring the mediating effects of attentional control, self-esteem, cognitive reappraisal, and the need to belong in the relationship between self-construal and cognitive biases. Through Structural Equation Modeling (SEM), we tested these mediating roles and depicted our findings in a comprehensive manner. Fig 2 graphically represents the intricate relationship between self-construal and cognitive bias as mediated by variables like attentional control, self-esteem, cognitive reappraisal, and the need to belong.

The model demonstrated excellent fit to the data($\chi^2/df$ = 2.347; RMSEA = 0.042; IFI = 0.993; CFI = 0.993; TLI = 0.965), and the detailed findings are as follows:

IndSC was negatively associated with the need to belong (β = -.099, p< .01) but positively linked with attentional control(β = .224, p < .001), self-esteem (β = .378, p < .001), cognitive reappraisal(β = .207, p < .001), and API (β = .17, p < .001). There was no significant direct effect of IndSC on ANI (β = .055, p = .136).

InterSC showed a positive association with cognitive reappraisal (β = .21, p< .001), the need to belong (β = .25, p< .001), and API (β = .154, p< .001), but was negatively related to attentional control (β = -.10, p < .05), and a marginally significant association with self-esteem (β = -.068, p = .061). The direct effect of InterSC on ANI (β = .052, p = .145) was not significant.

When both IndSC and InterSC were simultaneously included in the model, the direct effects of both IndSC (β = .17, p < .001) and InterSC (β = .154, p < .001) on API remained significant, while their direct effects on ANI were insignificant. In other words, self-construal exerted a more pronounced effect on API than on ANI.

Detailed in Table 2 are the findings from our tests on the direct and indirect effects of self-construal on cognitive bias. Regarding the indirect effects, among the indirect effects

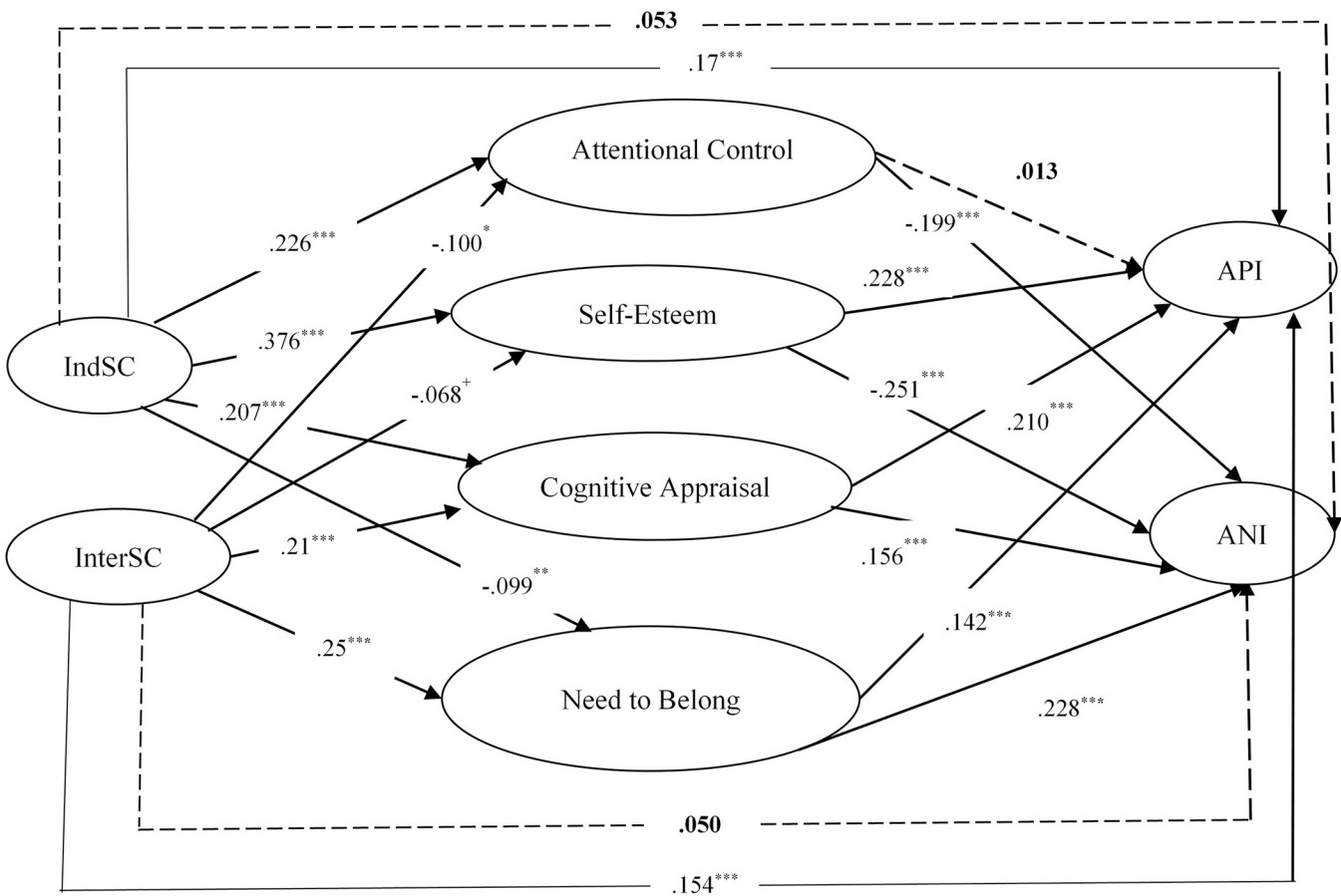

**Fig 2. Self-construal and cognitive bias, mediated by attentional control, self-esteem, cognitive reappraisal and need to belong in Chinese college students.** *Note*: API = attention to positive information; ANI = attention to negative information; IndSC = independent self-construal; InterSC = interdependent self-construal. **$P$<.01, *** $P$<.001, +$P$<.10.

of IndSC on API (β = .119, p < .001), the most substantial effect emanated from self-esteem (β = .086, p < .001), followed by cognitive appraisal (β = .044, p < .001), with the need to belong ranking last and exerting a negative indirect effect (β = -.014, p < .05). In the case of the indirect effect of IndSC on ANI (β = -.13, p < .001), it was entirely mediated through self-esteem (β = -.094, p < .001), attentional control (β = -.044, p < .001), cognitive appraisal (β = .030, p < .001), and the need to belong (β = -.022, p < .05). Self-esteem exhibited the most substantial effect among these variables. Notably, except for cognitive reappraisal, the remaining three variables displayed negative effects in the indirect effects of IndSC on ANI.

The indirect effects of InterSC on API (β = .064, p < .01) stemmed from cognitive appraisal (β = .044, p < .001) and the need to belong (β = .036, p < .001). Similarly, the indirect effect of InterSC on ANI (β = .124, p < .001) was entirely mediated through cognitive appraisal (β = .030, p < .001) and the need to belong (β = .057, p < .001). This underscored the complex pathways through which self-construal influences cognitive biases.

## 5. Discussion

Based on Markus and Kitayama's theory of self-construal and the Biopsychosocial Model, our findings indicate that self-construal, as a cultural factor (social dimension), is closely related to

**Table 2. Test of direct and indirect effects of self-construal and cognitive bias, mediated by attentional control, self-esteem, cognitive appraisal and need to belong in Chinese college students (*n* = 748).**

| Paths | Estimate | SE | Bootstrapping | | | | | | P |
|---|---|---|---|---|---|---|---|---|---|
| | | | Bias-Corrected 95%CI | | | Percentile 95%CI | | | |
| | | | Lower | Upper | P | Lower | Upper | | |
| **Standardized Direct Effects** | | | | | | | | | |
| IndSC→API | .170 | .043 | .089 | .258 | .001 | .086 | .255 | | .002 |
| IndSC→ANI | .055 | .040 | -.020 | .133 | .170 | -.022 | .131 | | .189 |
| InterSC→API | .154 | .041 | .074 | .234 | .001 | .075 | .235 | | .001 |
| InterSC→ANI | .052 | .042 | -.030 | .138 | .197 | -.031 | .136 | | .212 |
| **Standardized Indirect Effects** | | | | | | | | | |
| ***IndSC→API*** | .119 | .021 | .079 | .161 | .001 | .079 | .161 | | .001 |
| IndSC→attentional control→API | .003 | .009 | -.017 | .022 | .792 | -.017 | .022 | | .815 |
| IndSC→self-esteem→API | .086 | .014 | .038 | .091 | .001 | .038 | .091 | | .001 |
| IndSC→cognitive reappaisal→API | .044 | .009 | .017 | .053 | .000 | .015 | .050 | | .001 |
| IndSC→need to belong→API | -.014 | .005 | -.024 | -.002 | .010 | -.022 | -.001 | | .016 |
| ***IndSC→ANI*** | -.130 | .025 | -.180 | -.081 | .001 | -.179 | -.080 | | .001 |
| IndSC→attentional control→ANI | -.044 | .013 | -.071 | -.013 | .000 | -.064 | -.011 | | .001 |
| IndSC→self-esteem→ANI | -.094 | .014 | -.101 | -.045 | .001 | -.098 | -.043 | | .001 |
| IndSC→cognitive reappaisal→ANI | .030 | .008 | .011 | .042 | .001 | .011 | .041 | | .001 |
| IndSC→need to belong→ANI | -.022 | .008 | -.033 | -.004 | .013 | -.032 | -.003 | | .015 |
| ***InterSC→API*** | .064 | .019 | .027 | .101 | .003 | .027 | .100 | | .004 |
| InterSC→attentional control→API | -.001 | .005 | -.014 | .008 | .630 | -.011 | .010 | | .825 |
| InterSC→self-esteem→API | -.015 | .007 | -.029 | .000 | .058 | -.028 | .001 | | .075 |
| InterSC→cognitive reappaisal→API | .044 | .010 | .018 | .056 | .000 | .016 | .054 | | .001 |
| InterSC→need to belong→API | .036 | .009 | .011 | .047 | .001 | .010 | .045 | | .001 |
| ***InterSC→ANI*** | .124 | .025 | .077 | .174 | .001 | .078 | .175 | | .001 |
| InterSC→attentional control→ANI | .020 | .010 | .001 | .041 | .031 | .000 | .038 | | .053 |
| InterSC→self-esteem→ANI | .017 | .008 | -.001 | .031 | .060 | -.027 | .102 | | .075 |
| InterSC→cognitive reappaisal→ANI | .030 | .008 | .013 | .045 | .001 | .011 | .044 | | .001 |
| InterSC→need to belong→ANI | .057 | .010 | .025 | .066 | .001 | .025 | .065 | | .001 |

*Note*: API = attention to positive information; ANI = attention to negative information; IndSC = independent self-construal; InterSC = interdependent self-construal.

individual attention control (cognitive dimension), cognitive reappraisal (emotional regulation dimension), self-esteem (personality dimension), and the need to belong (affective motivation dimension). This multidimensional analysis reveals how these different aspects interact with each other, collectively shaping cognitive biases. Particularly, we found that the need to belong and self-esteem play a crucial role in modulating cognitive biases, emphasizing the importance of considering individual social and psychological needs in mental health interventions.

## 5.1 Self-construal and cognitive bias

Our analysis revealed that individuals with a pronounced IndSC exhibited a positive cognitive bias, as evidenced by an increased API and a decreased ANI. This trend aligns with the theoretical expectations that individuals who value personal autonomy and independence are more likely to engage with information that reinforces a positive self-view, potentially as a strategy to maintain self-esteem and reduce cognitive dissonance [79]. Such a disposition may

be protective in the context of mental health, mitigating the impact of negative information that could otherwise contribute to stress and anxiety.

Conversely, our study found that students with a strong InterSC attended to both positive and negative information, suggesting a cognitive style that is attuned to maintaining social harmony and relationships. This balanced cognitive bias may reflect an adaptive strategy within a collectivist framework, where attention to a broad spectrum of social cues is essential for fulfilling relational roles and obligations. These findings resonate with Yang et al.'s research, which suggested that individuals with a collectivist orientation may enhance their focus on positive social cues to facilitate reconnection following experiences of social exclusion [82]. This pattern underscores the salience of relationship dynamics in the cognitive processing of individuals with an InterSC orientation and highlights the importance of considering relational factors in mental health within collectivist cultures.

Our findings suggest that cognitive biases are personalized expressions influenced by cultural and individual self-construal orientations, which advocate for culturally adapted mental health interventions. For individuals with an IndSC orientation, mental health strategies might include enhancing positive information processing, whereas for those with an InterSC orientation, interventions could focus on managing the emotional impact of broad social information processing.

## 5.2 Mediating factors

Our study identified several key factors that mediate the relationship between self-construal and cognitive bias.

**5.2.1 Attentional control.** Attentional control exhibited a differential association with IndSC and InterSC, supporting our hypothesis about its mediating role. We found that IndSC correlates with heightened attentional control, suggesting an inclination towards efficient cognitive processing and possibly a reduced susceptibility to stress and anxiety. InterSC, however, was linked to a reduced attentional control, hinting at a potential vulnerability to distraction by social stimuli, which could influence emotional well-being. The finding aligns with the notion that individuals with a relational-contextual cognitive style, typical of InterSC, might overly focus on relationship dynamics, potentially at the cost of cognitive resources needed for other tasks [26]. The negative relationship between attentional control and ANI is particularly revealing; it suggests a protective mechanism that diverts focus from potentially distressing stimuli.

In the broader context of cultural psychology, these results underline the importance of attentional mechanisms in shaping the cognitive effects of self-construal. They suggest that enhancing attentional control might be a key strategy in psychological interventions tailored for individuals with different cultural backgrounds and self-construal orientations.

**5.2.2 Self-esteem.** In line with theoretical predictions, our findings revealed that students with a high degree of IndSC reported elevated levels of self-esteem. This heightened self-esteem correlated with an increased API and decreased ANI, resonating with Lam's research on the linkage between independent orientation, positive self-perception, and mental health outcomes [83]. Similarly, Kwan et al. have delineated the mediating role of global self-esteem between independence and life satisfaction [84].

Contrasting with expectations, our data did not show self-esteem mediating the effects of InterSC on API and ANI. This divergence from prior studies (e.g., [84] may stem from our focus on individual self-esteem without a corresponding emphasis on relational self-esteem, which can be particularly salient in collectivist cultures like China, where interpersonal relationships heavily influence self-worth [85]. These insights suggest that self-construals may

impact cognitive biases through the lens of self-esteem in culturally specific ways, and need for a deeper investigation into the multifaceted nature of self-esteem. Nevertheless, our results indirectly suggest that individuals with InterSC may derive their self-esteem from harmonious interpersonal relations, while those with IndSC may base it on individual accomplishments, in line with Hannover et al. and Hu et al. [33,37].

Our study contributes to the broader discourse on self-esteem in collectivist societies, proposing that individual accomplishments may elevate self-esteem and consequently skew cognitive processing towards positive stimuli. On the other hand, individuals with a strong InterSC may derive self-esteem from social harmony, which might not translate into a significant impact on cognitive biases as hypothesized. This nuanced understanding calls for a more differentiated approach in mental health interventions, taking into account the individual and relational aspects of self-esteem.

**5.2.3 need to belong.** research supports the hypothesis that the need to belong plays a significant role in mediating the impact of self-construal on cognitive biases. Students with a stronger IndSC orientation exhibited a negative influence on cognitive bias when a pronounced need for social connectedness was present, corroborating the findings of Gabriel et al. [10], which suggest that reminders of social relationships may not bolster the cognitive bias toward positive information for those who value independence. Furthermore, He et al. [86] noted that individuals who highly value independence often place less reliance on others for their social satisfaction, underscoring a potential strategy to enhance positive cognitive biases in mental health interventions: by decreasing the emphasis on belonging for those with an IndSC orientation.

Conversely, InterSC individuals showed an increased need to belong, which was associated with a broader attentional bias encompassing both positive and negative stimuli. This pattern of results suggests that for those who emphasize relational connectedness, a broadened scope of information processing is essential for managing social interactions, a concept supported by the work of DeWall and Richman [87] and Baumeister and Leary [88]. This increased sensitivity to social cues is a double-edged sword: it may contribute to well-being in supportive social environments but could also lead to heightened vulnerability in contexts of social exclusion or conflict.

The findings indicate that the need to belong may have mood-congruent effects for those with a strong InterSC, echoing previous research which suggests that fostering interpersonal connections can be beneficial for such individuals [10]. Therefore, interventions that skillfully leverage the need to belong could enhance positive cognitive biases while also managing negative biases, which is particularly relevant for mental health strategies aimed at individuals with a strong InterSC.

**5.2.4 Cognitive reappraisal.** Our results suggest that both IndSC and InterSC are associated with the use of cognitive reappraisal strategies among Chinese college students, albeit with different outcomes. This finding aligns partially with our hypotheses and highlights the complex role of cognitive reappraisal in emotional regulation.

Students with an IndSC orientation, who typically value personal goals and achievements, may use cognitive reappraisal to focus on positive aspects of a situation, thus enhancing their API. This utilization of cognitive reappraisal can be understood as a strategy to maintain self-consistency and reduce cognitive dissonance, which is consistent with the self-construal theory and Western literature on the subject [9]. Conversely, those with an InterSC orientation appear to use cognitive reappraisal to navigate social relationships and meet situational demands, which may lead to a more balanced attention to both positive and negative information.

The cultural context of contemporary China, where individualism and collectivism coexist [89], may contribute to the broad application of cognitive reappraisal across different self-construals. This cultural blend reflects a unique adaptability in emotional regulation

strategies, which can be a valuable asset in managing the complex emotional landscape of modern society [90].

Interestingly, cognitive reappraisal was found to enhance both API and ANI, suggesting that it can be a versatile tool in emotion regulation, potentially amplifying both positive and negative cognitive biases. This versatility emphasizes the importance of context in the application of cognitive reappraisal strategies. For instance, positive reappraisal of ego-focused emotions may boost API for those with high IndSC, while those with high InterSC might experience increased ANI if they suppress these emotions in favor of maintaining social harmony [91].

Our findings underscore the importance of culturally informed interventions that consider individual differences in self-construal. Interventions could be tailored to help individuals with an IndSC to maximize the benefits of cognitive reappraisal for positive emotions, while assisting those with an InterSC in managing the potential drawbacks of focusing on negative social cues.

## 6. Conclusions

Our research provides comprehensive insights into cognitive biases among Chinese college students, delving into the complex relationships between self-construal, cognitive mechanisms, motivational factors, and emotion regulation strategies. We observed that self-construal plays a pivotal role in shaping cognitive bias. IndSC is associated with increased attention to positive information and reduced attention to negative information. In contrast, InterSC is linked to a balanced focus on both positive and negative stimuli. These patterns indicate the adaptability of individuals within the Chinese cultural context and underscore the importance of self-construal in cognitive processing.

Our findings also highlight the critical mediating roles of attentional control, self-esteem, cognitive reappraisal, and the need to belong in the relationship between self-construal and cognitive bias. Attentional control was seen to play a significant role in diverting attention away from negative stimuli, particularly for those with an IndSC orientation. Self-esteem emerged as a crucial factor, especially for IndSC individuals, correlating with enhanced positive cognitive bias and reduced negative bias. Cognitive reappraisal was used by both IndSC and InterSC individuals, albeit for different emotional regulation purposes, influencing their attention towards both positive and negative information. Lastly, the need to belong was found to have contrasting effects based on self-construal orientation, affecting cognitive biases towards both valences in complex ways.

Theoretical Contributions: This research builds upon Markus and Kitayama's self-construal theory, applying it specifically to the context of Chinese college students. It extends the theory by exploring how independent and interdependent self-construals influence cognitive biases, moving beyond broad cultural categorizations to individual-level analysis within a collectivist society.

Practical Implications: For mental health interventions in collectivist cultures like China, our study suggests the importance of tailoring strategies to individual self-construal orientations. Additionally, understanding the mediating roles of attentional control, self-esteem, cognitive reappraisal, and the need to belong can guide the development of more nuanced and effective mental health interventions. For example, interventions could focus on enhancing attentional control and self-esteem in individuals with an IndSC, while emphasizing social skills and relational aspects for those with an InterSC.

In summary, our study enriches the understanding of the interplay between self-construal, cognitive biases, and mediating psychological factors. It underscores the need for culturally sensitive and individualized approaches in mental health practices, considering the diverse

cognitive and emotional styles influenced by self-construal within a rapidly evolving cultural landscape.

## 6.1. Limitations and future directions

Our study, anchored in Markus and Kitayama's self-construal theory and the Biopsychosocial Model, offers significant insights into the interplay of self-construal, cognitive biases, and mental health. However, it is important to acknowledge certain limitations:

Direct Mental Health Measurements: The absence of direct measurements for specific mental health indicators, such as anxiety and depression, is a notable limitation. Although the Rosenberg Self-Esteem Scale helped explore self-esteem's relationship with mental health, future studies should include a broader range of direct mental health assessments for a more comprehensive understanding.

Causal Inferences: The cross-sectional nature of our study limits the ability to draw causal inferences. Future research employing longitudinal or experimental designs would provide a stronger foundation for establishing causal relationships.

Generalizability: Our findings, derived from Chinese college students, may not be fully applicable to other demographic or cultural groups. Expanding future studies to include more diverse populations would enhance the generalizability of the results.

Future research should also explore methodological improvements, such as the incorporation of objective measures alongside self-reports, to minimize response biases. Additionally, acknowledging the cultural diversity within China, it would be beneficial to investigate how regional and subcultural differences may impact self-construal and cognitive processing.

In summary, while our study contributes valuable perspectives to the field, these limitations highlight areas for future exploration and refinement. By addressing these aspects, subsequent research can deepen our understanding of the complex dynamics between culture, cognition, and mental health.

## Author Contributions

**Conceptualization:** Jing Li.

**Funding acquisition:** Jing Li.

**Investigation:** Sijia Liu, Hongsheng Peng, Liwu Tang.

**Methodology:** Sijia Liu, Hongsheng Peng, Liwu Tang.

**Supervision:** Jing Li.

**Writing – original draft:** Jing Li, Sijia Liu, Lin Yuan.

**Writing – review & editing:** Jing Li.

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
