## [Decision Letter · Decision Letter 0]

8 Jan 2024

PMEN-D-23-00041

Self-Construal and Attentional Biases in Cognitive Processing: Insights from Chinese College Students for Mental Health Contexts

PLOS Mental Health

Dear Dr. Li,

Thank you for submitting your manuscript to PLOS Mental Health. After careful consideration, we feel that it has merit but does not fully meet PLOS Mental Health’s publication criteria as it currently stands. Therefore, we invite you to submit a revised version of the manuscript that addresses the points raised during the review process.

We look forward to receiving your revised manuscript.

Kind regards,

Ansar Abbas

Academic Editor

PLOS Mental Health

Journal Requirements:

1. Please amend your detailed online Financial Disclosure statement. This is published with the article. It must therefore be completed in full sentences and contain the exact wording you wish to be published.

a) State the initials, alongside each funding source, of each author to receive each grant. For example: "This work was supported by the National Institutes of Health (####### to AM; ###### to CJ) and the National Science Foundation (###### to AM).

2. Please update your online Competing Interests statement. If you have no competing interests to declare, please state: “The authors have declared that no competing interests exist.”

3. Please provide separate figure files in .tif or .eps format only and ensure that all files are under our size limit of 10MB. You may leave the figure captions or legends in the manuscript.

For more information about how to convert your figure files please see our guidelines: https://journals.plos.org/digitalhealth/s/figures

4. Please include a separate legend for each figure in your manuscript.

Additional Editor Comments (if provided):

Reviewers' comments:

Reviewer's Responses to Questions

**Comments to the Author**

1. Does this manuscript meet PLOS Mental Health’s publication criteria? Is the manuscript technically sound, and do the data support the conclusions? The manuscript must describe methodologically and ethically rigorous research with conclusions that are appropriately drawn based on the data presented.

Reviewer #1: Partly

Reviewer #2: Yes

2. Has the statistical analysis been performed appropriately and rigorously?

Reviewer #1: Yes

Reviewer #2: Yes

3. Have the authors made all data underlying the findings in their manuscript fully available (please refer to the Data Availability Statement at the start of the manuscript PDF file)?

Reviewer #1: No

Reviewer #2: Yes

4. Is the manuscript presented in an intelligible fashion and written in standard English?

Reviewer #1: No

Reviewer #2: Yes

5. Review Comments to the Author

Reviewer #1: The manuscript contains bullets and numbers used unnecessarily. These must be removed and prepare the document adequately. title is written well. Data analyses are good. Study seems to be significant but needs minor revision before publication.

Reviewer #2: Dear Authors,

Thank you for submitting your manuscript titled "Self-Construal and Attentional Biases in Cognitive Processing: Insights from Chinese College Students for Mental Health Contexts" for review. Your work contributes significantly to cultural psychology by elucidating the relationship between self-construal and cognitive biases. However, there are several methodological and theoretical aspects that warrant further attention and clarification.

Methodological Considerations:

The study discusses the link between self-construal, cognitive biases, and mental health, yet it does not include direct measures of anxiety and depression. While the use of Rosenberg’s Self-Esteem Scale highlights the role of self-esteem in mental health outcomes, incorporating direct assessments of anxiety and depression would substantively strengthen the implications of your findings in the context of mental health.

Additionally, the manuscript omits details about whether participants were screened for other mental health pathologies that could affect their responses to the questionnaires. Including such screening processes would enhance the validity of your findings, ensuring that the observed cognitive biases are more directly related to self-construal and not influenced by other mental health conditions.

Your employment of Derryberry and Reed’s Attentional Control Scale for assessing attentional control is commendable. However, the study lacks a comprehensive cognitive or executive function screening. Given the potential impact of these functions on cognitive biases and mental health, the inclusion of additional cognitive assessments could provide a more holistic understanding of the underlying mechanisms. A sole reliance on one scale might not adequately analyze attentional control in its entirety.

The study's dependence on self-reported measures could introduce biases such as social desirability. These limitations should be acknowledged and discussed in the manuscript to provide a more balanced view of the results.

Discussion:

Grounded in Markus and Kitayama’s self-construal theory, your study provides important insights. However, exploring additional theoretical frameworks could further illuminate the complex dynamics between cognitive biases and mental health. This approach would offer a broader perspective and deepen the understanding of these relationships.

6. PLOS authors have the option to publish the peer review history of their article (what does this mean?). If published, this will include your full peer review and any attached files.

**Do you want your identity to be public for this peer review?** For information about this choice, including consent withdrawal, please see our Privacy Policy.

Reviewer #1: No

Reviewer #2: **Yes: **Juan Felipe Cardona

---

## [Decision Letter · Decision Letter 1]

29 Feb 2024

Self-Construal and Attentional Biases in Cognitive Processing: Insights from Chinese College Students for Mental Health Contexts

PMEN-D-23-00041R1

Dear Dr. Li,

We are pleased to inform you that your manuscript 'Self-Construal and Attentional Biases in Cognitive Processing: Insights from Chinese College Students for Mental Health Contexts' has been provisionally accepted for publication in PLOS Mental Health.

Best regards,

Ansar Abbas

Academic Editor

PLOS Mental Health

Reviewer Comments (if any, and for reference):

Reviewer's Responses to Questions

**Comments to the Author**

1. If the authors have adequately addressed your comments raised in a previous round of review and you feel that this manuscript is now acceptable for publication, you may indicate that here to bypass the “Comments to the Author” section, enter your conflict of interest statement in the “Confidential to Editor” section, and submit your "Accept" recommendation.

Reviewer #2: All comments have been addressed

2. Does this manuscript meet PLOS Mental Health’s publication criteria? Is the manuscript technically sound, and do the data support the conclusions? The manuscript must describe methodologically and ethically rigorous research with conclusions that are appropriately drawn based on the data presented.

Reviewer #2: Yes

3. Has the statistical analysis been performed appropriately and rigorously?

Reviewer #2: Yes

4. Have the authors made all data underlying the findings in their manuscript fully available (please refer to the Data Availability Statement at the start of the manuscript PDF file)?

Reviewer #2: Yes

5. Is the manuscript presented in an intelligible fashion and written in standard English?

Reviewer #2: Yes

6. Review Comments to the Author

Reviewer #2: Dear Authors,

I am pleased to inform you that after careful review of your revised manuscript titled "Self-Construal and Attentional Biases in Cognitive Processing: Insights from Chinese College Students for Mental Health Contexts," all requested revisions have been satisfactorily addressed. Your commitment to enhancing the manuscript based on the feedback provided is commendable. The adjustments have significantly improved the clarity and impact of your research findings.

7. PLOS authors have the option to publish the peer review history of their article (what does this mean?). If published, this will include your full peer review and any attached files.

**Do you want your identity to be public for this peer review?** For information about this choice, including consent withdrawal, please see our Privacy Policy.

Reviewer #2: No
